# Transcriptome Analysis of Retinal and Choroidal Pathologies in Aged BALB/c Mice Following Systemic Neonatal Murine Cytomegalovirus Infection

**DOI:** 10.3390/ijms24054322

**Published:** 2023-02-21

**Authors:** Xinyan Zhang, Jinxian Xu, Brendan Marshall, Zheng Dong, Yutao Liu, Diego G. Espinosa-Heidmann, Ming Zhang

**Affiliations:** 1Department of Cellular Biology and Anatomy, Medical College of Georgia, Augusta University, Augusta, GA 30912, USA; 2James and Jean Vision Discovery Institute, Medical College of Georgia, Augusta University, Augusta, GA 30912, USA; 3Charlie Norwood VA Medical Center, Augusta, GA 30904, USA; 4Center for Biotechnology and Genomic Medicine, Medical College of Georgia, Augusta University, Augusta, GA 30912, USA; 5Department of Ophthamology, Medical College of Georgia, Augusta University, Augusta, GA 30912, USA

**Keywords:** cytomegalovirus, age-related macular degeneration, RNA sequencing, latency, inflammation, degeneration

## Abstract

Our previous studies have shown that systemic neonatal murine cytomegalovirus (MCMV) infection of BALB/c mice spread to the eye with subsequent establishment of latency in choroid/RPE. In this study, RNA sequencing (RNA-Seq) analysis was used to determine the molecular genetic changes and pathways affected by ocular MCMV latency. MCMV (50 pfu per mouse) or medium as control were injected intra-peritoneally (i.p.) into BALB/c mice at <3 days after birth. At 18 months post injection, the mice were euthanized, and the eyes were collected and prepared for RNA-Seq. Compared to three uninfected control eyes, we identified 321 differentially expressed genes (DEGs) in six infected eyes. Using the QIAGEN Ingenuity Pathway Analysis (QIAGEN IPA), we identified 17 affected canonical pathways, 10 of which function in neuroretinal signaling, with the majority of DEGs being downregulated, while 7 pathways function in upregulated immune/inflammatory responses. Retinal and epithelial cell death pathways involving both apoptosis and necroptosis were also activated. MCMV ocular latency is associated with upregulation of immune and inflammatory responses and downregulation of multiple neuroretinal signaling pathways. Cell death signaling pathways are also activated and contribute to the degeneration of photoreceptors, RPE, and choroidal capillaries.

## 1. Introduction

Human cytomegalovirus (HCMV) is a common virus which infects 40 to 80% of individuals in the human population [1]. The virus is usually acquired during early life when the innate and adaptive immune systems are not fully mature [2], with the eye being one of the major target organs. Thus, the incidence of HCMV chorioretinitis is reported to be 25% in infants with symptomatic congenital HCMV infection [3,4,5,6]. Furthermore, the choroid/RPE may be a site of HCMV latency, since recent studies in our laboratory of ocular tissue from human cadavers have revealed that HCMV DNA was present in 17% (4 of 24) of choroid/RPE samples [7].

In order to investigate the effects of lifelong infection with cytomegalovirus, we established an in vivo mouse model in which systemic neonatal murine cytomegalovirus (MCMV) infection of BALB/c mice spreads to the eye, with subsequent establishment of latency in the choroid/RPE and the development of retinal and choroidal pathologies that appear in aged, infected mice, including deposits at both basal and apical aspects of the RPE (basal lamina deposits and subretinal deposits), together with degeneration of the choriocapillaris, RPE, and photoreceptors [8]. These pathologies exhibit some features of human age-related macular degeneration (AMD), which is characterized in its early stages by lipoprotein deposits at the basal and apical aspects of the RPE and in its advanced forms by CNV [9,10] or geographic atrophy (GA) of the outer retinal tissue, RPE, and choriocapillaris [11,12,13]. AMD is a complex, multifactorial, progressive disease and a leading cause of severe, permanent vision loss in older individuals [9,10,14]. Although the exact events which contribute to the development of AMD remain uncertain, studies have implicated various immunological and inflammatory mechanisms [15,16,17]. Many studies support correlations between oxidative stress, persistent low-grade inflammation, and AMD pathogenesis [18,19,20,21,22,23]. Inflammation and oxidative stress are correlated physiological processes [18,24,25,26,27]. Superoxide radicals are formed by NADPH oxidase from activated immune cells during inflammatory reactions, while oxidants are activators of the NF-κB pathway triggering inflammation/immune response [28].

Our studies have indicated that MCMV most likely remains latent in eyes of aged mice, since replicating virus was never recovered and expression of virus genes *m83* and *m54*, which, respectively, encode a virion protein expressed late in the viral replication cycle [29] and a virus DNA polymerase, [30] was not detected in any of the eyes of the infected mice. However, expression of several other virus genes was detected in the eyes of aged mice [8]. The majority of these genes function in either immune-modulation, such as *m04* and *m138* [31], or inhibition of cell death, including *m36*, *m38.5*, *m41,* and *m45* [32], which could facilitate the establishment of ocular virus latency. Furthermore, other genes with the potential to stimulate immune responses, such as *IE1*, *m80,* and *m18,* were also expressed in the eyes of latently infected, aged mice [8]. IE1 can trigger a proinflammatory host transcriptional response via a STAT1-dependent mechanism [33,34], and *m80* encodes a MCMV assembly protein protease [35], while *m18* encodes a protein which is an antigenic peptide recognized by CD8 T cells [36] and also drives expression of the RAE-1 family of NKG2D ligands leading to subsequent activation of NK cells [37]. Therefore, activation of these genes could stimulate an inflammatory/immune response, which could, in turn, induce retinal and choroidal pathologies observed in infected aged BALB/c mice. In this study, we analyzed eye samples from aged, infected mice and age-matched uninfected controls by performing RNA sequencing (RNA-Seq) to identify molecular pathways involved in immune/inflammatory responses as well as others affected by ocular MCMV latency.

## 2. Results

### 2.1. SD-OCT (Spectral-Domain Optical Coherence Tomography) Analysis 

Six eyes from the MCMV latently infected mice at 18 months p.i. and three eyes from age-matched uninfected controls were used for sequencing analysis. Prior to removal of the eyes, SD-OCT examinations were performed, and retinal thickness was calculated using a Leica Envisu R2210 system (Bioptigen, Leica, Morrisville, NC, USA). Compared to age-matched uninfected control eyes, a significantly reduced retinal thickness was observed in all six eyes of the infected mice (Figure 1B) as previously reported [8], while severe photoreceptor (PRC) degeneration, including disappearance of the entire outer nuclear layer (ONL) in some areas, was observed in three of six eyes (Figure 1A). In addition, one choroidal neovascularization (CNV)-like lesion was observed in one of these three eyes with severe PRC degeneration (not shown).

### 2.2. Identification of Differentially Expressed Genes

Each sample generated 80 to 107 million paired reads with 100% passing filter (PF) clusters, >96% perfect barcode, >95% bases with Q30, and >39 mean quality scores.

The number of genes (≥1 normalized count) expressed in each sample ranged from 12,875 to 14,230 with a cutoff of |fold change| ≥ 2 and q ≤ 0.05 used to identify differentially expressed genes (DEGs). Compared to the uninfected control eyes (samples c1, c2, c3), 321 DEGs (208 downregulated and 113 upregulated, Appendix A) were identified in the six virus latently infected eyes. The top 15 upregulated and downregulated genes are listed in Table 1, together with the fold changes and FDR values. Among virus-infected eyes, 48 DEGs (38 downregulated and 10 upregulated, Appendix A) were identified in three eyes with severe retinal degeneration (samples h1, h2, h3), compared to three infected eyes with milder retinal degeneration (samples m1, m2, m3). A heat map for all nine samples is shown in Figure 1D.

### 2.3. Validation of Differentially Expressed Genes 

RNA-seq transcriptome data were validated by qRT-PCR analysis and expression levels of eight genes were analyzed in three control and six virus-infected samples. As shown in Figure 1C, similar expression trends were observed by both qRT-PCR analysis and RNA-Seq for all eight genes. A correlation (R2) of 0.8904 was observed between the log2FC of ΔCT values derived by qRT-PCR analysis and the log_2_FC value derived by RNA-seq analysis.

### 2.4. Comparison of Infected and Uninfected Eyes 

#### 2.4.1. Analysis of Canonical Pathways

QIAGEN IPA analysis was used to identify involvement of canonical pathways with a *p*-value of less than 10^−2^. These pathways, together with the differentially expressed genes in each canonical pathway, are listed in Table 2. Among a total of 17 canonical pathways identified, 10 function in neuroretinal signaling, with the majority of differentiated regulated genes involved in these pathways being downregulated in infected eyes. As shown in Figure 2, for example, all 22 differentially expressed genes linked to the Phototransduction Pathway of rod and cone cells, were downregulated. These include genes involved in cAMP and PKA-mediated signaling, which participate in the modulation of presynaptic GABA release [38] and are themselves regulated by Relaxin signaling in a biphasic manner [39]. cAMP signaling also plays a role in regulating glutamatergic transmission via phosphorylation of certain ionotropic glutamate receptors [40] through dopamine-DARPP32 feedback. Since DARPP-32 is localized in horizontal cells, amacrine cells, and Müller glial cells, this may result in the disfunction of these important retinal cell types. Expression of genes involved in RNA splicing was also downregulated in infected eyes, consistent with previous results which have shown that RNA processing defects are associated with many diseases of the neuron, including retinal degeneration [41].

Other pathways affected include the Endothelin-1 (ET-1) pathway, which was also downregulated, and G beta gamma signaling. The neuropeptide ET-1 is expressed in multiple retinal cell types, including RPE, photoreceptors, the inner plexiform layer, and ganglion cell layer [42] and functions in neuromodulation and neurotransmission [42,43]. However, ET-1 could also activate ET receptors in the retinal and choroidal vasculature, suggesting an important role in regulating in situ blood flow [42]. G beta gamma signaling plays a critical role in rod cell function in low light conditions [44], as well as neuronal CREB signaling, which is one of the major regulators of neurotrophin responses [45,46], and G-protein coupled receptor signaling. 

In contrast, the transcription of genes involved in several immune response pathways was upregulated. These included pathways involved in NFAT-regulated dendritic cell maturation, phagosome formation, and communication between innate and adaptive immune cells. Upregulated gene transcription was also observed in the GP6 signaling pathway, which functions in platelet activation and thrombus formation [47]. The Tec kinase signaling pathway, which regulates lymphocyte development, activation, and differentiation [48], as well as in the opioid signaling pathway, which is critically involved in many physiological processes including neuroprotection and immune response [49,50,51]. 

#### 2.4.2. Analysis of Diseases and Functions

The QIAGEN IPA analysis also permitted the categorization of differentially expressed genes according to known disease associations and function as listed below. 

Ophthalmic Disease, Organismal Injury, and Visual System Function. Many differentially expressed genes were noted in sections of this category. As shown in Table 3, differential expression of genes implicated in retinal degeneration, including degeneration of the photoreceptor, rod, and outer segments of the cone cell, was detected with z scores greater than 2. Seven differentially expressed genes were noted with a z score of 1.568. In addition, differential expression of genes required for maintenance of retinal cell and photoreceptor function and quantity were noted with z scores less than −2.

2.Cell Death and Survival. As shown in Table 4, many differentially expressed genes implicated in the death of retinal cells and epithelial tissues, either by apoptosis or necrosis, were detected. TUNEL assays and Western blots were used to confirm this observation. As shown in Figure 3A, many TUNEL-positive cells were observed in the choroid (indicated by white arrows), RPE layer (indicated by arrow heads), and outer nuclear layer (indicated by red arrows) in virus-infected eyes, while in contrast, only a few TUNEL-positive cells were observed in the outer nuclear layer of eyes from age-matched uninfected mice. Eyes of MCMV neonatally infected mice, as well as the eyes of the age-matched, uninfected controls, were also analyzed by Western blotting. As shown in Figure 3B, MCMV infection was associated with increased production of cleaved caspase 3, RIP3, MLKL, and decreased production of rhodopsin, indicating that both apoptosis and necroptosis may contribute to cell death.

3.Cellular Movement and Immune Cell Trafficking. As shown in Table 5, significant changes in the transcription of genes involved in migration, infiltration, and activation of multiple immune cell types were observed in infected eyes. Immunostaining was used to confirm these observations. As shown in Figure 4, accumulation of Iba1 positive macrophage/microglia and GFAP positive Müller cells/glia was observed in the subretinal space and outer nuclear layer of the eyes of aged, infected mice, but not in age-matched uninfected controls.

#### 2.4.3. Analysis of Upstream Regulators

Pathway analysis identified 36 upstream regulators of these differentially expressed genes with activation Z scores of greater than 1.5 and *p*-values of overlap less than 0.05 (Table 6). The majority of these upstream regulators are cytokines (IL1, IFN, IL17A, TNF, etc.) or transcription factors (STAT1, STAT3) that are involved in innate immunity/inflammation. Several growth factors (VEGF, TGF) were also identified as activated upstream regulators.

As shown in the summary graph (Figure 5), upstream regulators, such as IL1, OSM, IL17A, and STAT1, stimulate activation and migration of immune cells, tissue degeneration, and antigen presentation.

### 2.5. Comparison of Infected Eyes with and without Severe Retinal Degeneration

Compared to infected eyes without severe retinal degeneration, we identified 48 DEGs in three infected eyes with severe retinal degeneration (Appendix A). As shown in Table 7, the majority of differentially regulated genes were involved in pathways related to neuroretinal signaling, cell death, and retinal degeneration. In contrast, no significant differences were detected in immune response pathways, such as those involved in migration, infiltration, or activation of immune cells, between infected eyes with and without severe retinal degeneration.

## 3. Discussion

The data presented here complements our earlier studies on lifelong MCMV latency in the retina. These studies demonstrated that systemic MCMV infection of newborn mice resulted in ocular pathology later in life which exhibited some similarities to AMD [8]. Although no infectious virus could be isolated from the retina during latent ocular infection, many gross pathological changes in the retinal architecture were observed. The transcriptional data presented here provides a theoretical framework for understanding retinal and choroidal pathologies in latently infected eyes and demonstrates widespread alterations in the expression of genes critical for maintaining retinal form and function. Thus, systemic virus infection of neonates results in disruption of normal ocular gene expression patterns even in advanced age. 

The studies presented herein support the idea that cytomegalovirus ocular latency could be associated with in situ inflammation. Transcription of many inflammatory molecules was elevated, and pathway analysis indicated that several inflammatory pathways, including IL1, STAT1, IL17A, and OSM, were activated with activation Z scores greater than 2 and *p*-values of overlap less than 0.05. These inflammatory molecules may activate immune cells including macrophages/microglia and could also induce degeneration of photoreceptor, RPE, or choriocapillaris, either directly or indirectly, via activation of immune cells. Our results also demonstrate that multiple cell death pathways, including caspase-3 dependent apoptosis, necrosis, and necroptosis are activated and contribute to degeneration of photoreceptors, RPE, and choroidal capillaries, although future studies are still needed to determine which pathways contribute to which aspects of retinal degeneration.

Our previous studies have shown that during systemic neonatal MCMV infection of 129S1/SvImJ (129S1) mice [7], MCMV spreads to the eye with subsequent establishment of latency in the choroid/RPE. Unlike the neonatal infection of BALB/c mice, only a few MCMV genes are expressed, and no remarkable retinal or choroidal pathology, such as deposits, degeneration of the choriocapillaris, RPE, or photoreceptors, was observed [7]. BALB/c mice, which are susceptible to light damage due to a lack of melanin [52,53], exhibited photoreceptor degeneration (infiltrating cells, loss of outer segments, and decreased retinal thickness) in aged, uninfected control mice in both our own previous studies [8], as well as those of Bell and colleagues [54]. Thus, melanin in the RPE may prevent light-induced retinal damage by absorbing most of the light passing through the pupil and helping to scavenge free radicals [53,55,56], thereby protecting photoreceptors from oxidative stress [57,58]. Melanin might also prevent toxicity through its anti-oxidative function [59] and could play a role in protecting choroidal blood vessels from light damage [60]. Therefore, the susceptibility of BALB/c mice to light damage [61,62,63,64] may be due to an accumulation of reactive oxygen species (ROS) and subsequent oxidative stress and inflammation [65,66]. Previous studies have suggested that oxidative stress mediates the initial activation of viral gene expression during cytomegalovirus latency [67,68]. Following light damage, oxidative stress in MCMV latently infected BALB/c mice could activate expression of ocular virus genes, which in turn might promote production of inflammatory/angiogenic factors, thereby facilitating development of retinal and choroidal pathologies. 

The choroid/RPE may be a site of HCMV latency since we have shown that HCMV DNA is present in some human choroid/RPE samples [7]. However, whether HCMV ocular latency contributes to human AMD remains to be determined. AMD is a complex multifactorial disease, and the majority of risk factors, including genetics, environmental insults, and age-related issues, are linked to the induction of oxidative stress, which could activate expression of ocular HCMV genes, resulting in the production of inflammatory/angiogenic factors and thereby facilitating development of an AMD-like pathology.

## 4. Materials and Methods

### 4.1. Cells and Virus

MCMV strain K181 was originally provided by Dr. Edward Morcarski, Emory University, Atlanta, GA. The virus was prepared from the salivary glands of MCMV-infected immunosuppressed BALB/c mice, and virus stocks were titered on monolayers of mouse embryo fibroblast (MEF) cells, as described previously [69]. A fresh aliquot of virus stocks was thawed and diluted to the appropriate concentration for each experiment.

### 4.2. Mice

We purchased breeding pairs of BALB/c mice from Jackson Laboratory (Bar Harbor, ME). All mice were given unrestricted access to food and water and were maintained on a 12 h light cycle alternating with a 12 h dark cycle. The breeding and treatment of animals in this study adhered to the Association for Research in Vision and Ophthalmology (ARVO) Statement for the Use of Animals in Ophthalmic and Vision Research and was approved by the Institutional Animal Care and Use Committee of Augusta University. The rd8 mutation was excluded by genotyping.

### 4.3. Antibodies

Anti-RPE65 (ab231782), anti-RIP3 (phosphor s232, ab195117), and anti-Rodopsin (ab5417) were purchased from abcam (Boston, MA, USA). Other antibodies used in this study were obtained from the following sources: anti-cleaved caspase 3 (#9664), anti-RIP1 (#3493), and anti-mouse β-actin (#3700) were purchased from Cell Signaling Technology, Inc. (Danvers, MA, USA). Mouse anti-GFAP (glial fibrillary acidic protein, specific for glia/Müller cells) was obtained from BD Biosciences (San Jose, CA, USA). Anti-mouse Iba1 was purchased from FUJIFILM WAKO Chemicals, U.S.A. Corporation (019-19741, Richmond, VA, USA). Goat anti-rabbit IgG HRP (sc2004) and rabbit anti-goat IgG HRP (sc2768) were sourced from Santa Cruz (Santa Cruz Biotechnology, CA, USA). Anti-Mouse IgG HRP Conjugate (W402B) was obtained from Promega (Madison WI, USA). Anti-mouse Alexa 488 and anti-rabbit Alexa 594 were obtained from Vector Laboratories, Inc. (Burlingame, CA, USA).

### 4.4. Experimental Design

A total amount of 50 pfu of MCMV or culture medium as control were injected into BALB/c mice within 3 days after birth via the intraperitoneal (i.p.) route. At 18 months post infection (p.i.), the mice were anesthetized, and spectral-domain (SD) optical coherence tomography (SD-OCT) was performed using the Bioptigen Spectral-Domain Ophthalmic Imaging System (En-visu R2200; Bioptigen, Morrisville, NC, USA). The OCT imaging protocol included averaged single B scan and volume intensity scans with images centered on the optic nerve head (1.4 mm × 1.4 mm, @0.0, 1000X100X4X1). Total retinal thickness was measured by manual assessment of retinal layers using InVivoVue™ Diver 2.4 software (Bioptigen) following the software introduction, as described previously [8]. The mice were euthanized, and the eyes were collected and prepared for RNA-Seq, immunofluorescence staining, Western blot, and real time RT-PCR (qRT-PCR) as described below.

### 4.5. RNA-Seq

#### 4.5.1. RNA Extraction and Quality Control

Six eyes from infected mice at eighteen months p.i., and three eyes from age-matched, uninfected controls were used for RNA sequencing. Following removal of the lens, total RNA was extracted from whole eyes using Trizol (Invitrogen, CA, USA) according to the manufacturer’s instructions, and all nine RNA samples were then treated with DNase to exclude genomic DNA contamination. RNA concentrations were obtained using a Nanodrop 2000c spectrophotometer (Thermo Scientific Inc., Waltham, MA, USA), while RNA integrity was assessed using an Agilent 2200 Tape station instrument (Agilent Technologies, Santa Clara, CA, USA. RNA Integrity Number (RIN) scores for the nine samples were 7.6, 7.7, 7.7, 7.8, 8.0, 8.2, 8.2, 8.8, and 8.9, respectively. One microgram of total RNA from each sample was used to prepare Ribo-Zero RNA-Seq libraries.

#### 4.5.2. Total RNA-Seq with Ribo-Zero Depletion

RNA-Seq libraries were prepared using the Illumina TruSeq Stranded Total RNA kit (Illumina, Inc., San Diego, CA, USA) according to the manufacturer’s protocol. Briefly, ribosomal RNA (rRNA) was removed using biotinylated, target-specific oligos combined with Ribo-Zero rRNA removal beads according to the Illumina Reference Guide (Illumina, San Diego, CA, USA). Following purification, the RNA was fragmented into small pieces using divalent cations at an elevated temperature. First strand cDNA synthesis was performed at 25 °C for 10 min, 42 °C for 15 min, and 70 °C for 15 min, using random hexamers and ProtoScript II Reverse Transcriptase (NEW ENGLAND BioLabs Inc.). For second strand cDNA synthesis, RNA templates were removed, and a second replacement strand was generated through the incorporation of dUTP (in place of dTTP, to maintain strand identity) and double-strand cDNA was generated. Blunt-ended cDNA was isolated from the second strand reaction mix using beads. The three ends of the cDNA were then adenylated, and the cDNA was ligated to indexing adaptors. PCR (15 cycles of 98 °C for 10 s, 60 °C for 30 s, and 72 °C for 30 s) was used to selectively enrich for DNA fragments with adapter molecules on both ends, and to amplify the amount of DNA in the library. Libraries were quantified and qualified using the D1000 Screen Tape on an Agilent 2200 Tape Station instrument and were normalized, pooled, and subjected to cluster and pair read sequencing for 150 cycles on a HiSeqX10 instrument (Illumina, Inc. San Diego, CA, USA), according to the manufacturer’s instructions. 

#### 4.5.3. Data Analysis Methods

Coding RNA data were analyzed by Rosalind (https://rosalind.onramp.bio/) (accessed on 21 September 2020), with a HyperScale architecture developed by OnRamp BioInformatics, Inc. (San Diego, CA, USA) [70]. Reads were trimmed using cutadapt [71]. Quality scores were assessed using FastQC [72], and reads were aligned to the Mus musculus genome build mm10 using STAR [73]. Individual sample reads were quantified using HTseq [74] and normalized via Relative Log Expression (RLE) using the DESeq2 R library [75]. Read distribution percentages, violin plots, identity heatmaps, and sample MDS plots were generated as part of the QC step using RSeQC [76]. DEseq2 was also used to calculate fold changes (FDs) and *p*-values. The significant gene set was selected with a q-value (*p*-value that has been adjusted for False Discovery Rate, FDR) of <0.05 threshold.

Using QIAGEN Ingenuity Pathway Analysis (QIAGEN IPA) (Ingenuity^®^ Systems, www.ingenuity.com) (accessed on 23 March 2021), we illustrated clustering of genes for the final heatmap of differentially expressed genes and performed functional analyses to identify relevant gene pathways and networks.

### 4.6. qRT-PCR

RNA-seq transcriptome data were validated by qRT-PCR analysis. All primer sequences used for qRT-PCR are shown in Appendix A. Genes were amplified in 20 µL reaction consisting of 10 µL 2×SYBR Mix (Bio-Rad), 0.2 µL of 20 pmol/µL primer mixture, and 1 µL cDNA, using CFX96TM Real Time PCR System (Bio-Rad). PCR conditions were as follows: 3 min at 94°C, followed by 40 cycles of 94 °C for 10 s, 60 °C for 20 s, and 72 °C for 30 s. All CT values were analyzed and normalized to β-actin using the method of 2–ΔΔCT.

### 4.7. Immunofluorescence Staining

Eyes (four eyes in each group) were embedded in OCT compound, frozen, and sectioned in a cryostat. Sections were then fixed with 4% paraformaldehyde for 15 min and stained by TUNEL assay (in Situ Cell Death Detection Kit, Fluorescein; Roche Diagnostics, Indianapolis, IN, USA) and/or for RPE-65, GFAP, Iba-1 as described previously [77,78].

### 4.8. Western Blot

Lenses were removed from harvested eyes and the remaining eye tissues homogenized in a lysis buffer containing protease inhibitors (Complete™ Lysis-M, Roche, Germany). Proteins were extracted as previously described [79,80], and equal amounts of protein were separated by 10% or 12% SDS-PAGE, followed by electroblotting onto polyvinylidene difluoride membranes (Amersham Biosciences, Amersham, UK). Following blocking with 5% nonfat dry milk for 1 h at room temperature, membranes were incubated overnight at 4 °C with primary antibody. Binding of HRP–conjugated secondary antibody was performed for 1 h at room temperature, and bands were visualized using chemiluminescence (ECL; GE Healthcare, Chicago, IL, USA). 

## Figures and Tables

**Figure 1 ijms-24-04322-f001:**
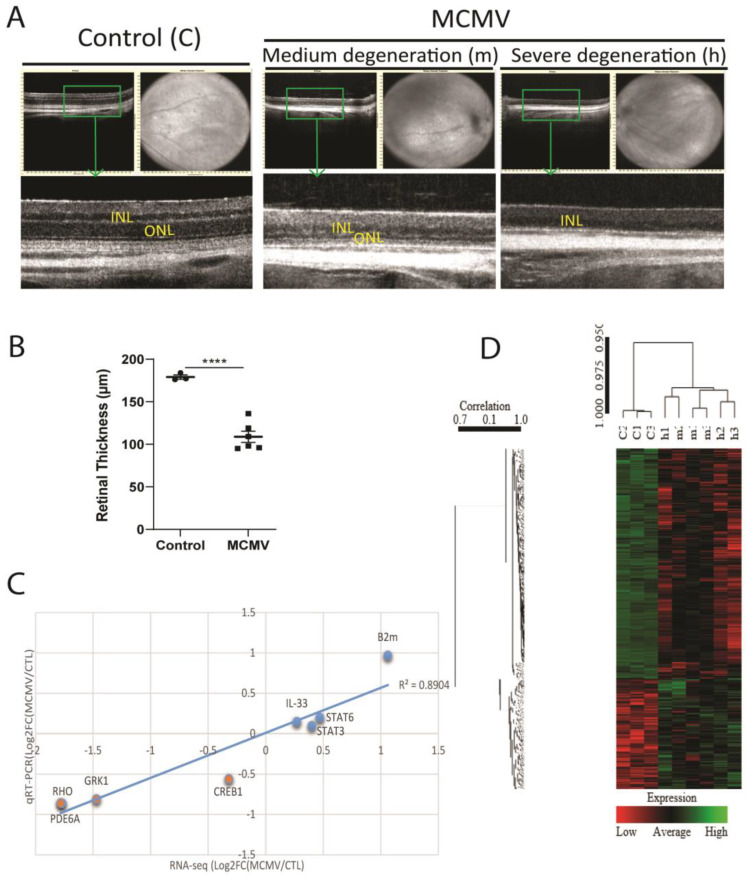
(**A**) Representative images of SD-OCT in eyes of latently infected mice with medium retinal degeneration (m) and severe retinal degeneration (h) at 18 months p.i., together with age-matched control eyes (c). (**B**) Retinal thickness by SD-OCT. **** *p* < 0.0001 using the Mann-Whitney test. (**C**) Validation of RNA-seq transcriptome data by qRT-PCR analysis of eight genes in three control and six virus-infected samples. (**D**) A Heat map for three control samples (c1, c2, c3) and six virus-infected samples (m1, m2, m3, h1, h2, h3).

**Figure 2 ijms-24-04322-f002:**
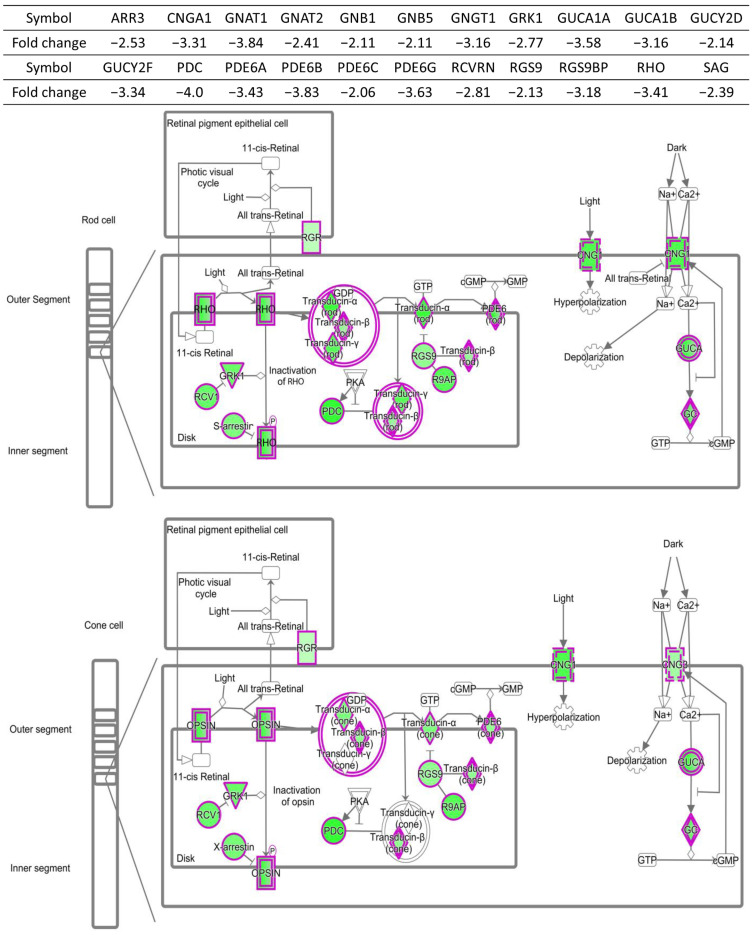
Downregulation of 22 differentially expressed genes linked to the phototransduction pathway of rod and cone cells.

**Figure 3 ijms-24-04322-f003:**
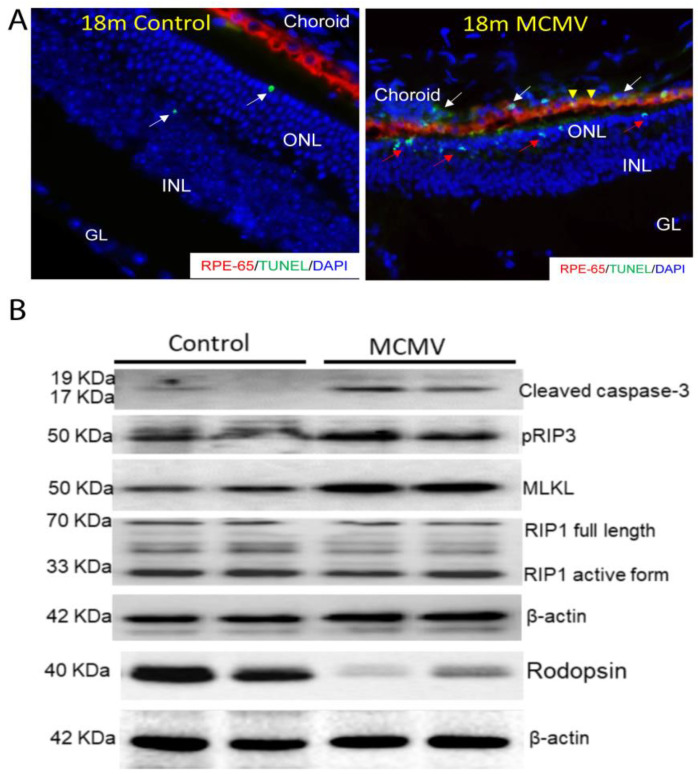
(**A**) Representative merged photomicrograph of staining for RPE-65 (red), TUNEL (green), and DAPI (blue) in eyes of latently infected mice at 18 months p.i. (MCMV) together with age-matched control eyes (control). (**B**) Western blot stained with antibodies against cleaved caspase 3, pRIP3, MLKL, RIP1, Rodopsin, and β-actin in eyes of latently infected mice at 18 months p.i. (MCMV) together with age-matched control eyes (control).

**Figure 4 ijms-24-04322-f004:**
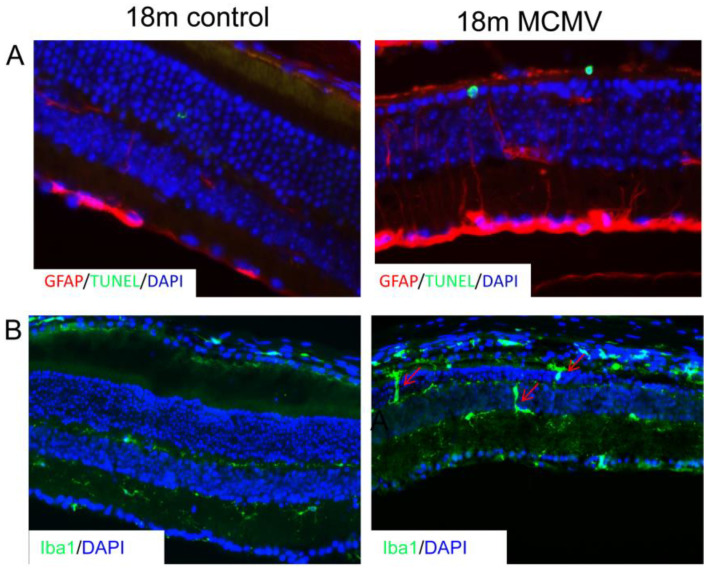
(**A**) Representative merged photomicrographs of stainings for (A) GFAP (red), TUNEL (green), and DAPI (blue) or (**B**) Iba-1 (red) and DAPI (blue) in eyes of latently infected mice at 18 months p.i. (MCMV) together with age-matched control eyes (control).

**Figure 5 ijms-24-04322-f005:**
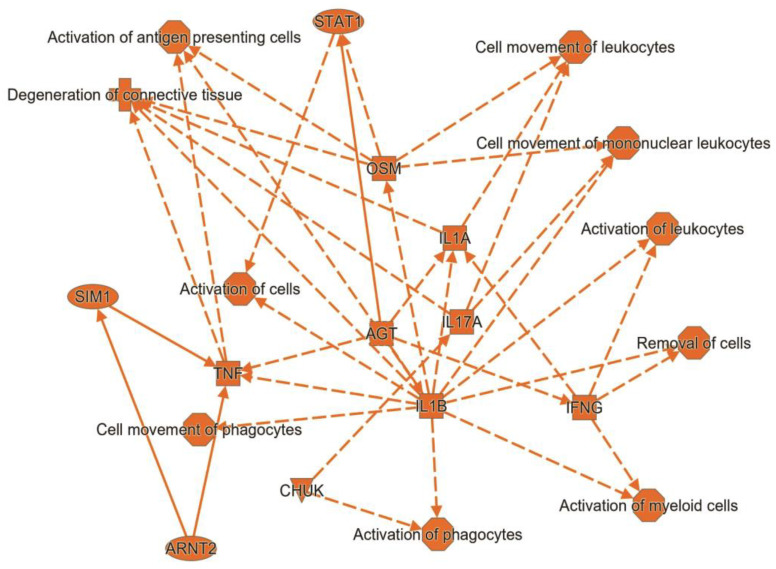
The summary graph of upstream regulators.

**Table 1 ijms-24-04322-t001:** Top mRNAs differentially expressed.

Gene Symbol	Gene Name	Fold Change	q Value
Gm14308	predicted gene 14308	11,223,350	3.56 × 10^−9^
Kdm5d	lysine (K)-specific demethylase 5D	2,210.25	1.17 ×10^−2^
Uty	ubiquitously transcribed tetratricopeptide repeat containing, Y-linked	968.76	3.60 × 10^−4^
Ddx3y	DEAD-box helicase 3 Y-linked	512	5.80 × 10^−5^
Eif2s3y	eukaryotic translation initiation factor 2, subunit 3, structural gene Y-linked	382.68	2.10 × 10^−5^
Krt16	keratin 16	38.85	4.24 × 10^−3^
S100a9	S100 calcium binding protein A9	28.44	1.62 × 10^−3^
Stfa2l1	stefin A2 like 1	25.63	3.49 × 10^−2^
Mmp13	matrix metallopeptidase 13	20.53	1.09 × 10^−2^
Sbsn	suprabasin	17.02	2.00 × 10^−2^
Ccr1	C-C motif chemokine receptor 1	11.08	1.96 × 10^−2^
Gm4951	predicted gene 4951	9.12	3.49 × 10^−7^
Gm4841	predicted gene 4841	8.51	4.24 × 10^−4^
Msr1	macrophage scavenger receptor 1	8.16	3.83 × 10^−3^
Nlrc5	NLR family CARD domain containing 5	5.81	1.83 × 10^−2^
Ccdc24	coiled-coil domain containing 24	−6.10	5.79 × 10^−4^
Pdzph1	PDZ and pleckstrin homology domains 1	−5.65	4.31 × 10^−4^
Apob	apolipoprotein B	−5.20	1.40 × 10^−5^
Sntg2	syntrophin gamma 2	−5.13	1.70 × 10^−5^
Glb1l3	galactosidase beta 1 like 3	−4.63	9.80 × 10^−5^
Alb	albumin	−4.56	9.31 × 10^−9^
Nr2e3	nuclear receptor subfamily 2 group E member 3	−4.14	5.00 × 10^−6^
Nrl	neural retina leucine zipper	−4.02	1.94 × 10^−3^
Pdc	phosducin	−4	8.00 × 10^−6^
Cabp4	calcium binding protein 4	−3.86	1.00 × 10^−6^
Wdr31	WD repeat domain 31	−3.86	2.71 × 10^−4^
Kcnv2	potassium voltage-gated channel modifier subfamily V member 2	−3.83	1.34 × 10^−4^
Pde6b	phosphodiesterase 6B	−3.83	6.23 × 10^−4^
Gnat1	G protein subunit alpha transducin 1	−3.83	6.81 × 10^−4^
Kcnj14	potassium inwardly rectifying channel subfamily J member 14	−3.75	8.52 × 10^−4^

**Table 2 ijms-24-04322-t002:** Ingenuity canonical pathways.

Ingenuity Canonical Pathways	*p*-Value	Ratio
Phototransduction Pathway	2.51 × 10^−28^	0.415
Relaxin Signaling	6.31 × 10^−7^	0.08
cAMP-mediated signaling	4.57 ×10^−5^	0.0526
tRNA Splicing	2.34 × 10^−4^	0.116
Protein Kinase A Signaling	2.45 × 10^−4^	0.0376
Dendritic Cell Maturation	6.61 × 10^−4^	0.0492
Phagosome Formation	0.0012	0.056
Dopamine-DARPP32 Feedback in cAMP Signaling	0.0013	0.0491
Role of NFAT in Regulation of the Immune Response	0.0025	0.0442
Endothelin-1 Signaling	0.0032	0.0426
GP6 Signaling Pathway	0.0046	0.0504
Opioid Signaling Pathway	0.005	0.0364
G Beta Gamma Signaling	0.0051	0.0492
Tec Kinase Signaling	0.0056	0.0427
CREB Signaling in Neurons	0.0056	0.0386
Communication between Innate and Adaptive Immune Cells	0.0081	0.0521
G-Protein Coupled Receptor Signaling	0.0093	0.0331

**Table 3 ijms-24-04322-t003:** Ophthalmic disease and visual system function.

Diseases or Functions Annotation	*p*-Value	Activation z-Score	Number of Molecules Affected
Degeneration of photoreceptors	1.58 × 10^−27^	4.897	38
Degeneration of retinal cone cells	2.29 × 10^−9^	2.956	10
Degeneration of outer nuclear layer	6.12 × 10^−9^	3.138	10
Degeneration of photoreceptor outer segments	4.49 × 10^−8^	2.121	8
Degeneration of retinal rods	3.31 × 10^−7^	2.433	8
Function of retina	8.69 × 10^−31^	−2.947	34
Function of retinal cells	9.64 × 10^−31^	−2.786	33
Quantity of photoreceptors	1.18 × 10^−18^	−3.908	24
Quantity of retinal cells	6.24 × 10^−17^	−3.521	25
Function of photoreceptors	2.05 × 10^−14^	−2.786	13
Photoresponse	2.o × 10^−12^	−2.328	14
Quantity of retinal cone cells	6.12 × 10^−09^	−2.343	10
Function of retinal rods	4.73 × 10^−08^	−2.433	7

**Table 4 ijms-24-04322-t004:** Cell death and survival.

Diseases or Functions Annotation	*p*-Value	Activation z-Score	Number of Molecules Affected
Cell death of retinal cells	6.42 × 10^−9^	0.943	11
Cell death of photoreceptors	1.40 × 10^−7^	1.3	8
Apoptosis of photoreceptors	2.20 × 10^−5^	0.152	5
Apoptosis of retinal cells	6.68 × 10^−5^	0.152	6
Apoptosis	4.98 × 10^−4^	0.936	81
Necrosis	3.0 × 10^−4^	0.258	80
Cell survival	9.14 × 10^−4^	0.029	49
Necrosis of epithelial tissue	0.003	1.651	22

**Table 5 ijms-24-04322-t005:** Immune cells and glia.

Diseases or Functions Annotation	*p*-Value	Activation z-Score	Number of Molecules Affected
Activation of Müller glia	9.3 × 10^−7^		4
Activation of neuroglia	1.65 × 10^−5^	1.079	11
Cellular infiltration by leukocytes	2.21 × 10^−5^	2.729	22
Leukocyte migration	8.61 × 10^−5^	2.472	38
Cell movement of leukocytes	2.25 × 10^−4^	2.518	32
Activation of phagocytes	4.93 × 10^−4^	1.673	17
Cellular infiltration by lymphocytes	6.84 × 10^−4^	0.957	10
Activation of macrophages	0.001	1.375	12
Activation of antigen presenting cells	0.00193	1.638	14
Cell movement of phagocytes	0.0031	1.597	22

**Table 6 ijms-24-04322-t006:** Upstream regulators.

Upstream Regulator	Molecule Type	Activation z-Score	*p*-Value of Overlap	Number of Molecules Affected
TGFB1	growth factor	3.005	3.99 × 10^−6^	48
APP	other	2.375	8.44 × 10^−6^	30
IFNG	cytokine	4.033	8.72 × 10^−6^	40
OSM	cytokine	2.439	5.58 × 10^−5^	25
CKAP2L	other	1.633	6.31 × 10^−5^	6
IFNB1	cytokine	1.975	1.4 × 10^−4^	13
ELAVL1	other	2.961	3.84 × 10^−4^	10
CHUK	kinase	1.976	5.0 × 10^−4^	10
CSF2	cytokine	2.597	6.47 × 10^−4^	17
IL17RA	transmembrane receptor	2	8.51 × 10^−4^	4
SMAD3	transcription regulator	1.769	0.00116	11
IL1B	cytokine	2.615	0.00125	26
AGT	growth factor	2.937	0.00139	23
Vegf	group	2.35	0.00148	16
ARNT2	transcription regulator	1.667	0.00161	9
IKBKG	kinase	2.401	0.00214	7
SIM1	transcription regulator	1.667	0.00216	9
PTGER2	G-protein coupled receptor	2.433	0.0024	6
MRTFA	transcription regulator	1.664	0.00289	8
STAT3	transcription regulator	1.689	0.00416	17
ERK	group	2.42	0.00522	9
TNF	cytokine	1.946	0.00548	38
RARA	ligand-dependent nuclear receptor	1.706	0.00844	11
EGF	growth factor	2.296	0.00932	15
IL17A	cytokine	2.768	0.00933	9
FOXO3	transcription regulator	1.953	0.0153	10
PI3K (complex)	complex	1.812	0.0168	9
IL1A	cytokine	2.146	0.017	8
IFN-α/β	group	1.941	0.0222	4
TGFA	growth factor	1.98	0.0222	4
STAT1	transcription regulator	2.684	0.0225	10
BMP4	growth factor	1.902	0.0271	7
PDGF BB	complex	2.086	0.0314	9
PRL	cytokine	1.643	0.0335	8
PTGS2	enzyme	2.216	0.0359	6
CD3	complex	1.978	0.0408	14

**Table 7 ijms-24-04322-t007:** Infected eyes with severe retinal degeneration vs. infected eyes without severe retinal degeneration.

Ingenuity Canonical Pathways and Diseases or Functions	*p*-Value	Activation z-Score	Number of Molecules Affected
Phototransduction pathway	1.26 × 10^−20^	N/A	11
Relaxin signaling	2.69 × 10^−3^	N/A	3
tRNA Splicing	2.95 × 10^−3^	N/A	2
Wnt/Ca+ pathway	5.89 × 10^−3^	N/A	2
Protein Kinase A signaling	6.17 × 10^−3^	N/A	4
cAMP-mediated signaling	8.51 × 10^−3^	N/A	3
Cell death of retinal cells	1.22 × 10^−10^	0.655	7
Cell death of photoreceptors	2.39 × 10^−8^	1.213	5
Retinal degeneration	3.6 × 10^−20^	2.766	18
Degeneration of photoreceptors	1.72 × 10^−11^	2.2	8

## Data Availability

The data presented in this study are included in this published article. All the data can be shared upon request by email.

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
