# Peer review of "Transcriptome Analysis of Retinal and Choroidal Pathologies in Aged BALB/c Mice Following Systemic Neonatal Murine Cytomegalovirus Infection"

_ijms, 2023, doi:10.3390/ijms24054322_

Round 1

Reviewer 1 Report

This is an interesting manuscript that determined transcriptome changes in retinas of a mouse model in which systemic neonatal murine cytomegalo virus (MCMV ) infection of BALB/c mice spreads to the eye, with subsequent establishment of latency in the choroid/RPE and development of retinal and choroidal pathologies resembling some features of human age-related macular degeneration (AMD). The data were clean and comprehensive. Data analysis and interpretation were appropriate, which provided novel mechanisms to understand retinal pathology in this clinic-relevant mouse model developed by the authors. I only have one minor suggestions: Fig.2, up panel is for Rod cells. Opsin in the rod is usually called as rhodopsin.

Fig.2, up panel is for Rod cells. Opsin in the rod is usually called as rhodopsin. Please make the change. Also, please modify up and lower panels as much as possible so that readers can easily tell the two panels are different. 

Author Response

We thank the reviewer for your very thorough and insightful comments.  We have listed your comments and questions below and our responses are provided immediately following each comment and/or question. Changes within the body of the manuscript are indicated by red font.

This is an interesting manuscript that determined transcriptome changes in retinas of a mouse model in which systemic neonatal murine cytomegalovirus (MCMV ) infection of BALB/c mice spreads to the eye, with subsequent establishment of latency in the choroid/RPE and development of retinal and choroidal pathologies resembling some features of human age-related macular degeneration (AMD). The data were clean and comprehensive. Data analysis and interpretation were appropriate, which provided novel mechanisms to understand retinal pathology in this clinic-relevant mouse model developed by the authors. I only have one minor suggestions: Fig.2, up panel is for Rod cells. Opsin in the rod is usually called as rhodopsin. Fig.2, up panel is for Rod cells. Opsin in the rod is usually called as rhodopsin. Please make the change. Also, please modify up and lower panels as much as possible so that readers can easily tell the two panels are different. 

Response: We agree with the reviewer and have replaced “opsin” in the upper panel with “RHO”. We have also modified the upper and lower panels for easier reading.

Reviewer 2 Report

Zhang et al. realized a very interesting article describing the “Transcriptome analysis of retinal and choroidal pathologies in aged BALB/c mice following systemic neonatal murine cytomegalovirus infection”. I consider the manuscript very interesting but, at the same time, I suggest several revisions needed to improve the reliability and the completeness of the paper:

.The “Introduction” sections should be more updated and improved. Mainly in the first part, when the authors discuss about RPE and degeneration of choriocapillaris, I suggest adding data related to the involvement of oxidative stress in relationship with immune response, also focusing on vascular components. The recent PMID: 32877751, PMID: 30523548, PMID: 36490268 and PMID: 36290689 could represent a substrate able to enforce the role of considered cellular mechanisms.

·  In my opinion, some figures could be shifted to Supplementary Materials.

· Are the experiments realized at least in triplicate?

·Why did authors use the FDR correction instead of Bonferroni or more robust one?

.Finally, manuscript requires important English revisions and typos correction.

Author Response

We thank the reviewer for your very thorough and insightful comments.  We have listed your comments and questions below and our responses are provided immediately following each comment and/or question. Changes within the body of the manuscript are indicated by red font.

Zhang et al. realized a very interesting article describing the “Transcriptome analysis of retinal and choroidal pathologies in aged BALB/c mice following systemic neonatal murine cytomegalovirus infection”. I consider the manuscript very interesting but, at the same time, I suggest several revisions needed to improve the reliability and the completeness of the paper:

The “Introduction” sections should be more updated and improved. Mainly in the first part, when the authors discuss about RPE and degeneration of choriocapillaris, I suggest adding data related to the involvement of oxidative stress in relationship with immune response, also focusing on vascular components. The recent PMID: 32877751, PMID: 30523548, PMID: 36490268 and PMID: 36290689 could represent a substrate able to enforce the role of considered cellular mechanisms.

      Response:  We have moved the second paragraph in the Discussion section (line 246 to 254 in old version) to the Introduction (line 52-60 in updated version).  The following paragraph has been inserted together with citations of recent papers as recommended by the reviewers.

 Inflammation and oxidative stress are correlated physiological processes.[18, 24-27] Superoxide radicals are formed by NADPH oxidase of activated immune cells during inflammatory reactions, while oxidants are activators of the NF-κB pathway triggering inflammation/immune responses. [28]

Are the experiments realized at least in triplicate?

Response: Yes. Each group for RNA-seq contained 3 eye samples and each group for  immunofluorescence staining or Western blotting contained 4 eye samples.

Why did authors use the FDR correction instead of Bonferroni or more robust one?

.     Response: This is a very good question. False Discovery Rate (FDR) adjusted p values (i.e., q value) are  widely used in gene expression and second-generation sequencing analysis. Although Bonferroni corrected p values may be more stringent through application of a lower cutoff, the expression of many genes in the human or mouse  genome is not completely independent of other genes. As a result, FDR-adjusted p values are considered to be appropriate for this experimental design which has been used in our previous expression analyses (PMID 32392310, 29860458, 22229401312, 26567786, and 24003086). Therefore, we applied FDR-adjusted p values in the expression analyses described in this manuscript.

Finally, manuscript requires important English revisions and typos correction.

Response: We have corrected the typos revised the grammar.

Round 2

Reviewer 2 Report

This manuscript can be accepted in present form